# Mechanical Protective Ventilation: New Paradigms in Thoracic Surgery

**DOI:** 10.3390/jcm14051674

**Published:** 2025-03-01

**Authors:** Mert Canbaz, Emre Şentürk, Mert Şentürk

**Affiliations:** 1Department of Anesthesiology and Reanimation, Istanbul Faculty of Medicine, University of Istanbul, 34093 Istanbul, Turkey; mcanbaz@ku.edu.tr; 2Department of Anesthesiology, Acibadem Atasehir Hospital, 34758 Istanbul, Turkey; dr.emresentrk@gmail.com; 3Department of Anesthesiology and Reanimation, School of Medicine, Acibadem University, 34758 Istanbul, Turkey

**Keywords:** one-lung ventilation, thoracic anesthesia, ventilator-associated lung injury

## Abstract

One-lung ventilation (OLV) in thoracic anesthesia poses dual challenges: preventing hypoxemia and minimizing ventilator-associated lung injury (VALI). Advances such as fiberoptic bronchoscopy and improved anesthetic techniques have reduced hypoxemia, yet optimal management strategies remain uncertain. Protective ventilation, involving low tidal volumes (4–6 mL/kg), individualized PEEP, and selective alveolar recruitment maneuvers (ARM), seek to balance oxygenation and lung protection. However, questions persist regarding the ideal application of PEEP and ARM, as well as their integration into clinical practice. As for PEEP and ARM, further research is needed to address key questions and establish new guidelines.

## 1. Introduction

One-lung ventilation (OLV) is a specific and one of the most important challenges of thoracic anesthesia. One expected issue is hypoxia, as the continuing perfusion to the non-ventilated lung would lead to an increased ratio of pulmonary shunt. As a matter of fact, hypoxia was and still is a common problem during OLV; however, its incidence has decreased from approximately 25% in the 1970s to around 5% in 21st century [1].

Another challenge, which was underestimated/ignored in earlier times is “ventilator associated/induced lung injury” (VALI) and the consequent “postoperative pulmonary complications” (PPC).

The experience about VALI in ICU patients can be useful to deal with “one-lung ventilation associated lung injury” [2,3]. Protective lung ventilation (PLV) consists of a bundle of low tidal volume (TV), PEEP, and alveolar recruitment maneuver (ARM) to achieve a low driving pressure. Hoverer, the application of PLV during OLV is associated with a series of questions (e.g., how low the TV; how high the PEEP, etc). The concept of “open lung approach” (OLA) can help to solve some of these questions by “individualizing” the PEEP with a titration [4]. However, the beneficial effects of OLA during OLV also remain controversial.

This narrative review aims to focus on the changes in ventilatory approach with regard to challenging problems.

*The old and everlasting paradigm: dealing with hypoxemia*.

The pathway of the classical challenge of the thoracic anesthesia, i.e., hypoxemia, is straight forward:“The non-dependent lung is not ventilated, as the perfusion persists; so that the whole perfusion to this lung adds to intrapulmonary shunt, leading to hypoxemia”.

The recommendations were also straightforward, such as:○The “hypoxic pulmonary vasoconstriction” (HPV) should be protected○FiO_2_ should be switched to 1.0 routinely○The tidal volume of two lungs should be given to “one” (dependent) lung○No PEEP to the dependent lung, as the increased pressure can divert the perfusion to non-ventilated lung.

However, improved experience has led to radical changes in this knowledge in the following years.

Most importantly, it has been realized that increased shunt is only one of the factors causing hypoxemia. Very probably, the suboptimal position of the airway device (e.g., double-lumen tube, which was placed only with auscultation) is the most possible (or common) reason of hypoxemia, as it has been reported that the routine use of fiberoptic bronchoscopy (FOB) was associated with the decrease in the incidence of hypoxemia from 25% to 5% [5]. Another possible reason is the ventilation-perfusion (V/Q) mismatch within the ventilated lung, i.e., “upper” parts of the ventilated lung are ventilated better, as the perfusion is still better in the lower parts. This means the possible reasons of hypoxemia during OLV are (not exclusively): malposition of airway device, increased shunt (non-ventilated lung is still perfused), and V/Q mismatch within the ventilated lung. As the historical approach has focused only on shunt, recent strategies suggest to use FOB (to prevent malposition) and different ventilation techniques to decrease the V/Q mismatch.It is not only the increased use of FOB but also advancements in anesthetic and surgical techniques and drugs that have played a role in the decrease in the frequency of hypoxemia. Yet, it has to be underlined that it is still an important challenge. Therefore, an anesthetist has to be familiar with the theory of hypoxic pulmonary vasoconstriction (HPV):HPV (the name is self-explaining) denotes the physiological phenomenon whereby pulmonary arterioles undergo constriction in response to alveolar hypoxia, and it mitigates V/Q mismatch. It is characterized by a biphasic nature. Initially, there is a rapid onset within 100 msec, reaching a steady state within approximately 5–20 min. Subsequently, if alveolar hypoxemia persists, a more pronounced phase emerges around the 40th minute and a steady state within 60–120 min. If the second phase ensues, the HPV response persists for an extended duration. The most potent response occurs when approximately 20–40% of lung volume experiences an alveolar partial pressure of oxygen (P_A_O_2_) around 65–70 mmHg [6,7]. Volatile anesthetic agents impair HPV in a dose-dependent manner, but “modern” volatile anesthetics in their routine doses, with or without the combination of thoracic epidural anesthesia do not lead to a clinically relevant change in oxygenation [8].The switch of FiO_2_ to 1.0 is a rational instinct, only as a first step approach to “treat” the hypoxemia [9,10]; in other words, FiO_2_ of 1.0 makes sense if hypoxemia occurs, but the routine change at the initiation of OLV can be even harmful: First of all, this approach is irrational, if we assume that shunt leads to hypoxemia: “Hypoxemia as a result of increased shunt does not response to increase in FiO_2_”. One of the underestimated causes of hypoxemia during OLV is the V-Q mismatch (not shunt) within the dependent (ventilated) lung, and this is the part that benefits from the increase in FiO_2_. Another drawback of high FiO_2_ is the possibility of an “absorption atelectasis”, which is a more possible case in lateral decubitus position and the additional increase in pressure around the ventilated lung. Especially if no PEEP is applied as the historical guidelines suggest, the resulting atelectasis can worsen the hypoxemia. Furthermore, the lungs are highly vulnerable to oxygen toxicity due to their consistently high PO_2_, and hyperoxia elevates reactive oxygen species production, exceeding antioxidant capacity and leading to oxidative stress and potential tissue damage.About “high tidal volume and no PEEP”, it is now well known that the possible hazards overcome the benefits; not only regarding the PPC’s (which will be discussed in the next paragraphs) but also hypoxemia.

Actually, the change in this paradigm is not new; however, surveys report that in daily practice, a lot of patients undergoing OLV continue to receive tidal volume and PEEP levels outside of recommended thresholds; even worse is that the tidal volume is even higher in high-risk subgroups, potentially placing them at elevated risk for iatrogenic lung injury [11]. This fact leads to the other important challenge of OLV, namely VALI and PPC’s.

### The Relative New Paradigm: Dealing with Lung Injury

Despite the medical improvements in the perioperative period, postoperative pulmonary complications are one of the significant causes of morbidity and mortality in patients undergoing major surgery. Postoperative pulmonary complications encompass a wide range of respiratory-related conditions with different pathophysiologies occurring within the first week after surgery, including concepts such as atelectasis, lung infection, respiratory failure, and bronchospasm. The prevalence of atelectasis following thoracic surgery varies between 30 and 70%, and the mortality rates associated with this condition can reach up to 50% [12,13]. An observational prospective study has shown that regarding the ARISCAT-score, 87.4% of the patients undergoing thoracic surgery had moderate-to-high risk for PPCs, and the incidence of PPC was 45.7% [14].

Positive pressure mechanical ventilation has proven its efficacy during the polio outbreak in Copenhagen in 1952, significantly contributing to the reduction in mortality rates [15]. However, like most procedures in the medical literature, this intervention is not without risks and can lead to adverse outcomes. Shortly, different types of “traumas” have been shown to lead to lung injury [16]. Barotrauma caused by increased airway pressures and volutrauma resulting from volume changes due to pressure changes throughout the respiratory cycle cause significant transpulmonary pressure and mechanical stress and strain, leading to damage to alveolar epithelial cells. Atelectotrauma is named as the mechanism of damage arising from the repeated collapse and reopening of alveoli. These mechanisms not only damage alveolar structures but can also negatively affect vascular, epithelial, and endothelial cells leading to a significant pulmonary inflammatory response. These effects and their immunological consequences are referred to as biotrauma [16]. In the last decade, a fifth trauma has been defined, namely, ergotrauma, which adds the effects of PEEP and respiratory rate as elements of the so-called “mechanical power” of a possible lung injury [17].

Regarding the historical paradigm, “VALI appears exclusively in ICU, almost exclusively in ARDS-lungs”. Indeed, the majority of scientific knowledge and practical experience is about ARDS-lung in the ICU:The “sick” lung is more prone to injury,The duration of ventilation is longer than healthy lungs,Only a very small part of the lung is ventilated: the so-called “baby lung” of Gattinoni. Unfortunately, this “healthy” area is becoming more vulnerable to injury, as it receives the whole tidal volume [18].

It has come out that this was not correct, too: Mechanical ventilation is a non-physiologic process [19], and even the “healthy” lung undergoing only some hours of mechanical ventilation is under risk of lung injury [20]. For OLV, the risk of lung injury is even higher, as the ventilated “one” lung can be considered as an imitation of the “baby lung” [2]. Therefore, the ventilation strategy of OLV should deal with both challenges: It should prevent (and -if necessary- treat) hypoxemia and protect the lung against injury at the same time. Unfortunately, the optimal strategy is not defined yet, but there are some rational hints.

## 2. Protective Ventilation Strategies

Ventilation strategies applied can carry the risk of increased volutrauma or barotrauma at high volumes or high PEEP values, while low volumes or low PEEP values can cause damage through atelectrauma. Given these bilateral adverse effects, balanced protective ventilation strategies become of great importance [21,22].

“Protective” ventilation includes a bundle of several components; however, for all of these components, there are also unanswered questions (Table 1). Therefore, the current recommendation appears to be rational, but is exclusively empiric: low tidal volumes of 5–6 mL/kg based on ideal body weight, positive end-expiratory pressure of 5 cmH_2_O/or “individualized” PEEP, plateau pressure of less than 25 cmH_2_O, driving pressure of 13–14 cmH_2_O, and alveolar recruitment maneuvers (ARM) (only if necessary) [23,24,25]. All the components of protective ventilation are associated with unanswered questions; these components will be discussed in detail.

The theory of the protective ventilation is straight forward: ARM should open all alveoli, PEEP should keep them open, so that even the low TV would be enough to achieve a sufficient gas exchange; because the “optimal” PEEP would obtain a ventilation within the limits of the best compliance, a lower driving pressure would be required to achieve the ventilation. This approach would work not only to protect the lung but also prevent hypoxemia: All the alveoli should contribute to gas exchange; moreover, as the driving pressure is low and the PEEP optimal, the switch of perfusion to non-ventilated lung should be minimal. However, whether and how the theoretical background and practical outcome match is an ongoing question.

### 2.1. Tidal Volume

Previous studies and clinical applications have recommended the use of high tidal volumes during the intraoperative period to prevent adverse outcomes such as atelectasis [11,26]. Regarding the current knowledge, both experimental and clinical studies have found an association between the use of high tidal volumes and potential lung injury, considering high tidal volumes as a risk factor for clinically relevant pathophysiologies [27,28]. Low TV is therefore the least controversial component of the protective ventilation bundle. Still, it should be underlined that the potential benefits of even the “least controversial” component is dependent on an appropriate combination with other components.

Using low tidal volumes without adequate PEEP can lead to atelectasis, causing harm even in healthy lungs, with conditions such as bacterial translocation and pneumonia. This synergistic effect prevents atelectotrauma, alveolar distension, and barotrauma, reducing inflammation by preventing cyclic alveolar collapse and expansion. Protective ventilation strategies for one lung ventilation have not been clearly established, but approaches involving low tidal volume and PEEP have been recommended, with common principles such as ensuring oxygenation and preserving the lung [11].

Studies examining approaches to one lung ventilation primarily investigate the relationship between tidal volume changes and complications by maintaining other variables. In one study, a common PEEP value (5 cmH_2_O) was determined during upper abdominal surgery, and high and low tidal volume strategies were compared, with no difference found in postoperative lung complications [29]. During one lung ventilation at a common PEEP value of 5 cmH_2_O, different tidal volumes of 4 mL/kg, 6 mL/kg, and 8 mL/kg were compared, with the group using 6 mL/kg tidal volume showing lower postoperative complication rates and higher PaO_2_/FiO_2_ ratios [30]. A meta-analysis showed increased postoperative pulmonary complications with the use of low tidal volumes during one lung ventilation, but significant limitations were noted due to clinical heterogeneity and inclusion of a small number of articles [31]. Another systematic review demonstrated a significant relationship between low tidal volumes and postoperative pulmonary complications (OR, 0.40 [0.29–0.57]; *p* < 0.0001), although no differences in oxygenation, hospital stay, and mortality were found between low and traditional approaches in the same study [31].

### 2.2. Alveolar Recruitment Maneuver

There are also some discrepancies between the theory and practice of the effects of ARM: Studies showing the benefits of ARM have been followed by some other reporting its drawbacks, both reflecting the experience of daily practice.

While the majority of the studies in thoracic surgery have shown that recruitment maneuvers reduce both postoperative complications and improve oxygenation [32,33,34], some studies have shown that ARM has no effect or can be associated even with an increase in mortality [35,36].

Timing of the ARM can play an important role: it has been shown that recruitment maneuvers applied at the 30th minute of one lung ventilation did not increase arterial oxygen pressure, suggesting the timing of recruitment maneuvers may be crucial [37]. The meta-analysis related with ARMs has demonstrated that recruitment maneuvers during one lung ventilation are more beneficial compared to maneuvers applied before OLV initiation [38].

It appears that ARM can be beneficial, like the other components of the protective ventilation, only if it is combined with an appropriate contribution of other elements and with a rational timing.

The PROTHOR study aims to compare patients receiving high PEEP (10 cmH_2_O) and ARM with those receiving low PEEP (5 cmH_2_O) without ARM in terms of PPC’s; this trial recently finished [39]). As the biggest clinical trial in mechanical ventilation in thoracic anesthesia, the results can present a rational answer to existing controversies.

### 2.3. Pressures

#### 2.3.1. Driving Pressure

Driving pressure is used to interpret respiratory system compliance and can be calculated as the difference between plateau pressure and positive end-expiratory pressure. Static lung compliance is determined by the ratio of tidal volume to driving pressure. According to these formulations, driving pressure can also be calculated as the ratio of tidal volume to static compliance. Driving pressure has an inverse relationship with static compliance, meaning higher driving pressures reflect decreased lung compliance.

A driving pressure threshold of 15 cmH_2_O has been identified for mortality prediction in ARDS patients, with each 1-unit increase in driving pressure shown to increase mortality by 5% [40,41]. Evaluating planned protective ventilation strategies for surgical patients in a meta-analysis of 17 randomized controlled trials, Ary S Neto and colleagues found driving pressure to be a risk factor for developing pulmonary complications in multiple regression analyses (OR: 1.16, each 1 cmH_2_O increase) [42]. Subgroup analyses from the study indicated that higher PEEP values accompanying increased driving pressure increase the incidence of postoperative pulmonary complications (OR: 3.11, 95% CI), whereas higher PEEP values accompanying decreased driving pressure reduce it (OR: 0.19, 95% CI) [27,42,43].

#### 2.3.2. Positive End-Expiratory Pressure

Low PEEP levels (<5 cmH_2_O) are rarely used in clinical practice because they do not completely prevent alveolar collapse and have a weak effect on intrinsic PEEP. Reduced airway outflow during expiration can lead to residual pressure, known as intrinsic PEEP, which can reduce pulmonary dynamic compliance. Adjusting appropriate PEEP levels is necessary in patients undergoing one-lung ventilation or in lateral decubitus positions to balance increased intrinsic PEEP. It has been suggested that the ideal PEEP should be at least 80% of the intrinsic PEEP [44,45]. While low PEEP levels are less effective in raising oxygenation and less likely to raise intrathoracic pressure, studies have found higher rates of postoperative pulmonary complications and mortality [46]. On the other hand, high PEEP levels (>10 cmH_2_O) are beneficial in opening atelectatic alveolar areas and improving oxygenation but can lead to hemodynamic instability and decreased cardiac output due to venous return obstruction. Additionally, high PEEP levels have been shown to decrease lung compliance and increase the risk of barotrauma and bacterial translocation [47,48]. Thus, the effect of PEEP is two-fold: it can cause excessive alveolar expansion on one hand and alveolar collapse at small values on the other. Fixed PEEP values may not be optimal for every patient.

### 2.4. “Open Lung Approach” (OLA)

The drawbacks of “fixed” PEEP application can be solved with “individualization” of PEEP via different methods of titration. The “Open Lung approach” (OLA), proposed for ARDS patients, uses maximal alveolar recruitment followed by PEEP titration using the pressure-volume curve. The rationale of OLA is to target a decrease in driving pressure (DP), i.e., a lower pressure for the same tidal volume, or better to say a better static compliance (Figure 1). Several studies have found that OLA during thoracic surgery improved oxygenation, lung compliance, and transpulmonary pressure (PL), without significant hemodynamic effects, and indicated that PL is crucial in understanding lung and chest wall mechanics during one-lung ventilation [25,49,50,51].

The iPROVE-Network has investigated in a series of studies the effects of “individualized PEEP” determined by best dynamic compliance, accompanied by an alveolar recruitment maneuver (ARM) and low TV. The concept is based on opening the lung with an ARM, then checking the dynamic compliance in decremental PEEP values (i.e., lowest driving pressure for the same TV), followed by an ARM and low TV. Investigators have published three important studies about the effects of open-lung strategy with a “decremental PEEP titration” to adjust the PEEP with the best compliance. The first study has shown that this strategy has obtained an improvement in oxygenation during OLV [25]. In the second study, a lower incidence of PPC with open-lung approach compared to those reported in other thoracic surgery studies has been demonstrated [52]. In a recent RCT, the group has investigated the effects on PPC: it was shown that the individualized PEEP led to significantly better outcomes than fixed PEEP. Among 1308 patients, severe postoperative pulmonary complications were lower in the OLA group (6% versus 15%) than in the standard ventilation group [4].

A recent meta-analysis showed that individualized PEEP titration based on lung compliance reduces the risk of postoperative pulmonary complications during OLV and improves respiratory mechanics and oxygenation without affecting hemodynamic variables [53].

However, there are also some controversies about the effects of OLA: although Park et al. have shown in a previous RCT that a DP-guided ventilation strategy has reduced the incidence of postoperative pulmonary complications and pneumonia compared to a fixed PEEP [23], a more recent, larger study of the same group has not confirmed this result: decreased DP did not reduce pulmonary complications [35].

The question remains whether this method can be applied in daily practice [54], and especially in high-risk patients. Moreover, even a “best” PEEP (defined with any method) can only be “best” for a limited region of the lung: even with “best(?)” PEEP, the non-dependent areas tend to overinflate, and the dependent parts tend to have atelectasis [55]. Some studies have shown that PEEP levels corresponding to the expiratory limb bend point are more effective in improving oxygenation [56]. Although it can be useful in correlating with oxygenation index and determining optimum PEEP, it is not suitable for PEEP titration due to the need for real-time monitoring and intermittent blood gas sampling [45].

There are other different methods to help for the “titratio”n of the optimal PEEP [57,58,59,60,61] (Table 2). However, the feasibility of these methods in daily practice is technology and logistics-dependent and, therefore, questionable [57,58,59,60,61].

## 3. Future Directions and Clinical Implications

In the future, mechanical power (MP) and personalized ventilation strategies are expected to play a crucial role in optimizing OLV during thoracic surgery. MP and “ergotrauma” are relative recent concepts:

* The respiratory rate is taken also into account of VALI. 

* PEEP per se is considered as a part of MP.

These principles can again change the paradigm of protective ventilation.

In any case, it has become crucial that every physician applying mechanical ventilation has to be aware of the fact that she/he has to “protect” the lung against the possible hazards of this “unphysiologic” (RIP: Hedenstierna) approach (or better: against her/his hazardous application (RIP: Gattinoni).

The shift toward data-driven, real-time monitoring will allow for more precise adjustments to ventilator settings, reducing the risk of complications such as postoperative pulmonary dysfunction. As technology evolves, it is likely that these innovations will become routine in clinical practice, enhancing safety and recovery times for patients undergoing high-risk thoracic procedures.

## 4. Conclusions

In conclusion, OLV presents significant challenges in thoracic anesthesia, primarily in preventing hypoxemia and postoperative pulmonary complications. Recent advancements in protective ventilation strategies, including the use of low tidal volumes, optimal PEEP, and ARM when appropriate, have demonstrated promising outcomes in addressing both of these challenges. However, the effectiveness of these strategies relies on the precise application and individualized adjustment of each component: a TV of 4–6 mL/kg appears to be appropriate but is effective only with rational application of other components. Regarding PEEP and ARM, there are still important questions to be answered in order to set new paradigms. Notably, further investigation is needed to clarify the optimal PEEP levels and timing for ARM to fully understand their role in minimizing postoperative pulmonary complications. Thus, a tailored approach, the integration of these components, remains crucial to reducing postoperative pulmonary complications and improving recovery outcomes.

## Figures and Tables

**Figure 1 jcm-14-01674-f001:**
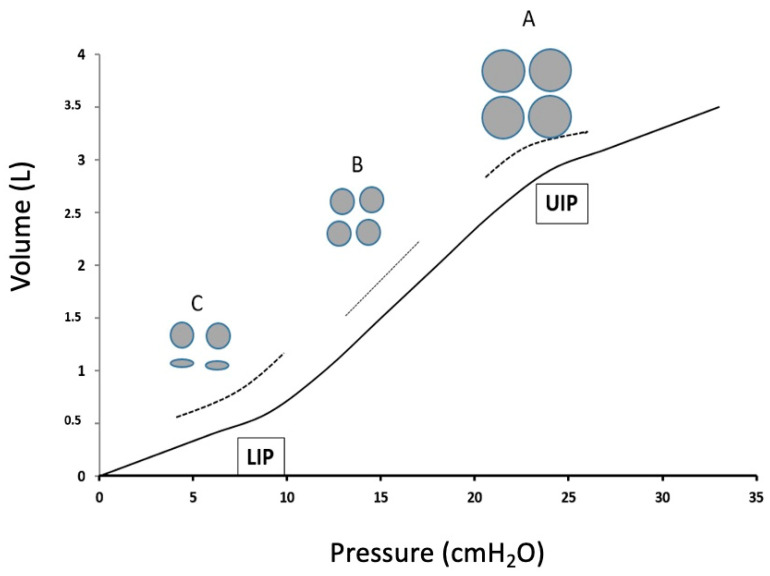
“Decremental PEEP titration” to achieve the lowest driving pressure according to the static compliance.The compliance curve has a sigmoid shape. The area with a steeper slope has a better compliance (i.e., less pressure is required for the same volume). The aim is to ventilate the lungs within this area, not below the lower inflection point (LIP), which would lead to atelectasis and cyclic opening-closing of the alveoli, and not above the upper inflection point (UIP), which would mean overinflation and lead to baro- or volutrauma. A: An ARM is applied to open all the atelectatic (and recruitable) alveoli, then the ventilation is started with a high PEEP. In this region, compliance is not expected to be high. B: A stepwise decrease in PEEP, and a stepwise increase in static compliance is expected. This increase would continue until some alveoli became atelectatic. C: When the compliance decreases again, some alveoli become atelectatic (below LIP). The PEEP level achieving the highest compliance is the individualized best PEEP; a new ARM must now be performed to open the atelectatic alveoli again, and the patient is ventilated with the PEEP of best compliance. Note: 1. This method should be performed ideally with the construction of P-V curve to identify LIP and UIP. 2. Distribution of ventilation within the ventilated lung is not homogenous; therefore, it should be kept in mind that even with the “individualized” PEEP, some (upper) regions are overinflated, as some (lower) are in risk of atelectasis.

**Table 1 jcm-14-01674-t001:** Components of “protective” lung ventilation.

Components	Purpose	Related Problems and Recommendations
**Low tidal volume (TV)**	- Prevent excessive inflation. - Prevent barotrauma, volutrauma, and biotrauma. - Reduced DP.	- How low should TV be? (Even 6 mL/kg of ideal body weight is approximately 10 mL/kg for ‘one’ lung. - For one-lung ventilation (OLV), TV of 4–6 mL/kg seems reasonable.
**Positive end-expiratory pressure (PEEP)**	- Preventing alveolar collapse and avoid cyclic opening-closing (atelectotrauma). - Improved compliance (?)	- What should the PEEP level be? (Higher PEEP can reopen atelectatic areas but may cause overdistention and hemodynamic complications.) - PEEP titration can be beneficial, but it is often challenging in everyday practice. Additionally, the ’optimal PEEP’ varies across different regions of the lung.
**Low driving pressure (DP)** **(Plateau pressure minus PEEP)**	- DP is clearly related, but not directly proportional to tidal volume. - Prevent overinflation. - Avoid barotrauma, volutrauma, and biotrauma. - Aim for improved compliance.	- How can I achieve the lowest DP while minimizing the impact on gas exchange? - Should I compensate with an increase in respiratory rate (RR)? - The combination of various components can vary from one patient to another.
**Alveolar recruitment maneuver (ARM)**	- Reopen the atelectatic alveoli for a more uniform ventilation/perfusion distribution.	- When should ARM be applied? - How should it be performed? - Is it beneficial for all patients? - It is not easy to determine whether ARM will be beneficial or potentially harmful, especially in cases of hemodynamic instability.
**Low mechanical power (MP)**	- The formula also incorporates PEEP and RR, highlighting that these factors can contribute to lung injury. - Prevent ergotrauma.	- Should I compensate for a low TV with an increased RR? In other words, which scenario poses a greater risk: low TV with a normal RR (risk of hypercarbia) or low TV with a higher RR (risk of excessive MP)? - What about using lower PEEP? - This is a less explored area in thoracic anesthesia and warrants further investigation.

**Table 2 jcm-14-01674-t002:** Some possible methods of PEEP titration. Historical methods like PEEP-FiO_2_ scale or “best DO2) are not included to the table; likewise, a CT-scan is impossible to apply during the operation in the operating room. (OR: operating room; VCV: volume-controlled ventilation; RR: respiratory rate).

Method	Principle	Comment/Concern
Identification of LIP and UIP on the P-V curve	PEEP to achieve the highest static compliance (see also Figure 1)	Static compliance (i.e., no inspiratory flow) is difficult to measure in daily practice. Titrated PEEP does not eliminate that some (lower) areas remain atelectatic, as some (upper) areas are overinflated.
Electric impedance tomography (EIT)	Visual detection of atelectasis. PEEP to avoid atelectasis.	Difficult in OR, esp. in thoracic patients.
Lung ultrasound	Visual detection of atelectasis	Less difficult than EIT, but still difficult during the operation. Subjective evaluation possible
Ventilatory stress index	Analysis of the slope of the pressure-time curve during VCV: PEEP to achieve a linear slope (not concave or convex)	Although being a very appropriate method, and easy at the same time, it is not used often in daily practice.
Transpulmonary pressure	Oesophageal pressure as the surrogate. The most direct way to measure the stress and strain	Specific motors necessary. Artifacts possible.
Mechanical power	Measurement takes the RR and PEEP also in account	Relative new method. Almost no experience for PEEP titration in OR.

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
