# Peer review of "Mechanical Protective Ventilation: New Paradigms in Thoracic Surgery"

_jcm, 2025, doi:10.3390/jcm14051674_

Round 1

Reviewer 1 Report

Comments and Suggestions for Authors

This is an interesting review dealing with the important topic of the appropriate application of mechanical ventilation (MV) during thoracic surgery with one-lung ventilation, in order to reduce postoperative pulmonary complications (PPC). The authors address the two main issues of MV concerning clinical practice, namely lung-protective ventilation (LPV) and open-lung approach (OLA). The historical concept of LPV consists of low Vt, low end-inspiratory pressure (Pplat) <30cmH2O and driving pressure (DP) of 12-14cmH2O. Recently, respiratory rate has emerged as an additional risk factor of ventilator-associated lung injury (VALI), contibuting to mechanical power transmitted to the lungs. On the other hand, the concept of OLA is defined as setting the PEEP level to maximize alveolar recruitment and keep alveoli open, avoiding cyclic atelectasis and atectrauma. It is used in combination with PLV to prevent VALI and improve patients' outcome. Individualization of PEEP is accomplished by various methods, among which stepwise alveolar recruitment maneuvers (ARM) have been used during thoracic surgery. Nowadays, there is little doubt that low Vt (4-6ml/KgIBW) during one-lung ventilation is associated with decreased rates of PPCs. On the contrary, the beneficial effects of OLA still remain controversial. 

As a general comment, I suggest to the authors to make this distinction clear, providing the above definitions and then build their review in a more structured way based on these two concepts.

Some further suggestions are quoted below:

Lines 57-60 are a little incomprehensible, they need to be explained. V/Q mismatch as a mechanism of hypoxemia should be also included here (lines 77-78)

Lines 71-73: you  mean "... volatile anesthetics ... inhibit HPV, but ...

In paragraph 4 (lines 74-83) oxygen toxicity should be mentioned

The components of LPV as summarized in table 1 should preferably be analysed one by one, using separate subtitles instead of using one subtitle (PRESSURE, line 164) to encompass all the elements.

Next, the OLA should be addressed, focusing on "optimal" PEEP (lines 189-206 and 221-228)

I also suggest a more detailed description of the reference 37, as this is an important RCT with a large number of patients that showed a significantly decreased rate of PPCs when individualised OLA  is combined with LPV. 

To enforce these data, I suggest the inclusion of a recent meta-analysis by Gu WJ et al, published in Crit Care 2025;Jan 15;29(1):27. 

Finally, a number of minor corrections should be made:

l. 38: induced instead of induces

l. 49: also instead of als

l. 58: suboptimal instead of supoptimal

l. 63: of HPV 

l. 71: PAO2 instead of PaO2

l. 75: application

l. 80: absorption 

l. 112: stress and strain

l. 114: of alveoli instead of alveolar structures

l. 116: omit the phrase "with the mechanical stress caused"

l. 118: The last decade, a fifth type of trauma has been proposed....

Table 1: risk of excessive MP instead of TV

l. 210: values

l. 225: computed 

Author Response

Response to Reviewer 1 Comments

Dear reviewer, first of all, thank you for your comments. In accordance with the suggestions you have made, we have made changes to the text, especially those related to OLA. We think that the article has been moved to a better level with your suggestions.

Response 1:

2O and driving pressure (DP) of 12-14cmH2O. Recently, respiratory rate has emerged as an additional risk factor of ventilator-associated lung injury (VALI), contibuting to mechanical power transmitted to the lungs. On the other hand, the concept of OLA is defined as setting the PEEP level to maximize alveolar recruitment and keep alveoli open, avoiding cyclic atelectasis and atectrauma. It is used in combination with PLV to prevent VALI and improve patients' outcome. Individualization of PEEP is accomplished by various methods, among which stepwise alveolar recruitment maneuvers (ARM) have been used during thoracic surgery. Nowadays, there is little doubt that low Vt (4-6ml/KgIBW) during one-lung ventilation is associated with decreased rates of PPCs. On the contrary, the beneficial effects of OLA still remain controversial. As a general comment, I suggest to the authors to make this distinction clear, providing the above definitions and then build their review in a more structured way based on these two concepts.>

Dear reviewer, thank you for the comments you have made. Changes have been made to the article in accordance with your suggestions, and a section about OLA has also been added to the article.

“Open Lung Approach” (OLA)

The drawbacks of “fixed” PEEP application can be solved with “individualization” of PEEP via different methods of titration. The "Open Lung approach" (OLA), proposed for ARDS patients, uses maximal alveolar recruitment followed by PEEP titration the pressure-volume curve. Several studies have found that OLA during thoracic surgery improved oxygenation, lung compliance, and transpulmonary pressure (PL), without significant hemodynamic effects, and indicated that PL is crucial in understanding lung and chest wall mechanics during one-lung ventilation [25,49-51].

The iPROVE-Network has investigated in a series of studies the effects of “individualised PEEP” determined by best dynamic compliance, accompanied by alveolar recruitment maneuver (ARM) and low TV. The concept is based on opening the lung with an ARM, then checking the dynamic compliance in decremental PEEP values (i.e. lowest driving pressure for the same TV) followed by an ARM and low TV. Investigators have published 3 important studies about the effects of open-lung strategy with a “decremental PEEP titration” to adjust the PEEP with the best compliance. The first study has shown that this strategy has obtained an improvement in oxygenation during OLV [25]. In the second study, a lower incidence of PPC with open-lung approach compared to those reported in other thoracic surgery studies has been demonstrated [52]. In a recent RCT, it was shown that he individualized PEEP led to significantly better outcomes than fixed PEEP. Among 1308 patients, severe postoperative pulmonary complications were lower in the OLA group (6%) than in the standard ventilation group [4]. 

A recent meta-analysis showed that individualized PEEP titration based on lung compliance reduces the risk of postoperative pulmonary complications during OLV, improves respiratory mechanics and oxygenation without affecting hemodynamic variables [53].

The rationale of OLA is to target a decrease of driving pressure (DP), i.e. a lower pressure for the same tidal volume, or better to say a better static compliance. (Figure 1). 

However, there is also some controversies about the effects of OLA: Although Park et al have shown in a previous RCT that a DP-guided ventilation strategy has reduced the incidence of postoperative pulmonary complications and pneumonia compared to a fixed PEEP [23], a more recent and bigger study of the same group has not confirmed this result: Decreased DP did not reduce pulmonary complications [35].

The question remains whether this method can be applied in daily practice, [54] and especially in high-risk patients. Moreover, even a “best” PEEP (defined with any method) can only be “best” for a limited region of the lung: even with “best(?)” PEEP, the non-dependent areas tend to an overinflation, and the dependent parts to atelectasis [55]. Some studies have shown that PEEP levels corresponding to the expiratory limb bend point are more effective in improving oxygenation [56]. Although it can be useful in correlating with oxygenation index and determining optimum PEEP, it is not suitable for PEEP titration due to the need for real-time monitoring and intermittent blood gas sampling [45].

There are other different methods to help for the “titratio”n of the optimal PEEP [57-61]. (Table 2). However,  the feasibility of these methods in daily practice is technology&logistics-dependend and therefore questionable [57-61].

Response 2:

Dear reviewer, thank you for your suggestion. In accordance with your suggestion, the relevant paragraph has been revised as follows:

Most importantly, it has been realized that increased shunt is only one of the reasons causing hypoxemia. Very probably, the suboptimal position of the airway device (e.g. double-lumen tube, which was placed only with auscultation) is the most possible (or: common) reason of hypoxemia, as it has been reported that the routine use of fiberoptic bronchoscopy (FOB) was associated with the decrease in the incidence of hypoxemia from 25 % to 5 % [5]. Another possible reason is the ventilation-perfusion (V/Q) mismatch within the ventilated lung, i.e. “upper” parts of the ventilated lung are ventilated better, as the perfusion is still better in the lower parts. This means, the possible reasons of hypoxemia during OLV are (not exclusively): malposition of airway device, increases shunt (non-ventilated lung is still ventilated) and V/Q mismatch within the ventilated lung. As the historical approach has focused only on shunt, recent strategies suggest to use FOB (to prevent malposition) and different ventilation techniques to decrease the V/Q shunt.

Response 3:

 inhibit HPV, but ...>

Dear reviewer, thank you for your suggestion. In accordance with your suggestion, the relevant section has been revised as follows:

Volatile anesthetic agents impair HPV in a dose-dependent manner, but “modern” volatile anesthetics in their routine doses, with or without the combination of thoracic epidural anesthesia do not lead to a clinically relevant change in oxygenation. (5)

Response 4:

Dear reviewer, thank you for your suggestion. In accordance with your suggestion, the following section has been added to the relevant section:

Furthermore, the lungs are highly vulnerable to oxygen toxicity due to their consistently high PO2, and hyperoxia elevates reactive oxygen species production, exceeding antioxidant capacity and leading to oxidative stress and potential tissue damage.

Response 5:

Dear reviewer, thank you for your suggestion. In accordance with your suggestion, the following headings have been added to the relevant section respectively;

Tidal volume;

Alveolar recruitment maneuver;

Pressures;

Driving pressure;

Positive End-Expiratory Pressure;

“Open Lung Approach” (OLA);

Response 6:

<Next, the OLA should be addressed, focusing on "optimal" PEEP (lines 189-206 and 221-228)>

Dear reviewer, thank you for your suggestion. In accordance with your suggestion, the following section has been added to the relevant section, and the bibliography has been updated:

“Open Lung Approach” (OLA)

The drawbacks of “fixed” PEEP application can be solved with “individualization” of PEEP via different methods of titration. The "Open Lung approach" (OLA), proposed for ARDS patients, uses maximal alveolar recruitment followed by PEEP titration the pressure-volume curve. Several studies have found that OLA during thoracic surgery improved oxygenation, lung compliance, and transpulmonary pressure (PL), without significant hemodynamic effects, and indicated that PL is crucial in understanding lung and chest wall mechanics during one-lung ventilation [25,49-51].

The iPROVE-Network has investigated in a series of studies the effects of “individualised PEEP” determined by best dynamic compliance, accompanied by alveolar recruitment maneuver (ARM) and low TV. The concept is based on opening the lung with an ARM, then checking the dynamic compliance in decremental PEEP values (i.e. lowest driving pressure for the same TV) followed by an ARM and low TV. Investigators have published 3 important studies about the effects of open-lung strategy with a “decremental PEEP titration” to adjust the PEEP with the best compliance. The first study has shown that this strategy has obtained an improvement in oxygenation during OLV [25]. In the second study, a lower incidence of PPC with open-lung approach compared to those reported in other thoracic surgery studies has been demonstrated [52]. In a recent RCT, it was shown that he individualized PEEP led to significantly better outcomes than fixed PEEP. Among 1308 patients, severe postoperative pulmonary complications were lower in the OLA group (6%) than in the standard ventilation group [4]. 

A recent meta-analysis showed that individualized PEEP titration based on lung compliance reduces the risk of postoperative pulmonary complications during OLV, improves respiratory mechanics and oxygenation without affecting hemodynamic variables [53].

The rationale of OLA is to target a decrease of driving pressure (DP), i.e. a lower pressure for the same tidal volume, or better to say a better static compliance. (Figure 1). 

However, there is also some controversies about the effects of OLA: Although Park et al have shown in a previous RCT that a DP-guided ventilation strategy has reduced the incidence of postoperative pulmonary complications and pneumonia compared to a fixed PEEP [23], a more recent and bigger study of the same group has not confirmed this result: Decreased DP did not reduce pulmonary complications [35].

The question remains whether this method can be applied in daily practice, [54] and especially in high-risk patients. Moreover, even a “best” PEEP (defined with any method) can only be “best” for a limited region of the lung: even with “best(?)” PEEP, the non-dependent areas tend to an overinflation, and the dependent parts to atelectasis [55]. Some studies have shown that PEEP levels corresponding to the expiratory limb bend point are more effective in improving oxygenation [56]. Although it can be useful in correlating with oxygenation index and determining optimum PEEP, it is not suitable for PEEP titration due to the need for real-time monitoring and intermittent blood gas sampling [45].

There are other different methods to help for the “titratio”n of the optimal PEEP [57-61]. (Table 2). However,  the feasibility of these methods in daily practice is technology&logistics-dependend and therefore questionable [57-61].

Response 7:

Dear reviewer, thank you for your suggestion. In accordance with your suggestion, the relevant section has been changed as follows.

In a recent RCT, the group has investigated the effects on PPC: it was shown that he individualized PEEP led to significantly better outcomes than fixed PEEP. Among 1308 patients, severe postoperative pulmonary complications were lower in the OLA group (6%) than in the standard ventilation group [4]. 

Response 8:

Dear reviewer, thank you for your suggestion. In accordance with your suggestion, the data related to our article from the relevant source have been added to the article in this way and the bibliography has been updated.

A recent meta-analysis showed that individualized PEEP titration based on lung compliance reduces the risk of postoperative pulmonary complications during OLV, improves respiratory mechanics and oxygenation without affecting hemodynamic variables [53].

  1. Gu, W.-J.; Zhao, F.-Z.; Piccioni, F.; Shi, R.; Si, X.; Chen, S.; Cecconi, M.; Yin, H.-Y. Individualized PEEP titration by lung compliance during one-lung ventilation: a meta-analysis. Critical Care 2025, 29, 27, doi:10.1186/s13054-024-05237-y.

Response 9:

<Finally, a number of minor corrections should be made:>

induced instead of induces>

Dear reviewer, thank you for your suggestion. In accordance with your suggestion, the relevant word has been corrected as follows:

…“ventilator associated/induced lung injury” (VALI) and the consequent “postoperative pulmonary complications” (PPC).

Response 10:

also instead of als>

Dear reviewer, thank you for your suggestion. In accordance with your suggestion, the relevant word has been corrected as follows:

Response 11:

…also straightforward, such as:

Response 12:

suboptimal instead of supoptimal>

Dear reviewer, thank you for your suggestion. In accordance with your suggestion, the relevant word has been corrected as follows:

…suboptimal position of the airway device (e.g. double-lumen tube, which was placed only with auscultation).

Response 13:

of HPV>

Dear reviewer, thank you for your suggestion. In accordance with your suggestion, the relevant word has been corrected as follows:

…of HPV:

Response 14:

AO2 instead of PaO2 >

Dear reviewer, thank you for your suggestion. In accordance with your suggestion, the relevant word has been corrected as follows:

…(PAO2) around 65-70 mmHg.

Response 15:

Dear reviewer, thank you for your suggestion. In accordance with your suggestion, the relevant word has been corrected as follows:

…change at the initiation of OLV can be even harmful:

Response 16:

absorption> 

Dear reviewer, thank you for your suggestion. In accordance with your suggestion, the relevant word has been corrected as follows:

…“absorption atelectasis”, which is a more possible case in lateral decubitus position and the additional increase of pressure around the ventilated lung.

Response 17:

and strain>

Dear reviewer, thank you for your suggestion. In accordance with your suggestion, the relevant word has been added.

… and strain, leading to damage to alveolar epithelial cells.

Response 18:

of alveoli instead of alveolar structures>

Dear reviewer, thank you for your suggestion. In accordance with your suggestion, the relevant word has been corrected as follows:

… alveoli.

Response 19:

Dear reviewer, thank you for your suggestion. In accordance with your suggestion, the relevant part has been deleted:

These mechanisms not only damage alveolar structures but can also negatively affect vascular, epithelial, and endothelial cells leading to a significant pulmonary inflammatory response.

Response 20:

Dear reviewer, thank you for your suggestion. In accordance with your suggestion, the relevant part has been corrected as follows:

The last decade, there is a fifth trauma has ben proposed namely ergotrauma, which adds the effects of PEEP and respiratory rate as elements of so-called “mechanical power” to a possible lung injury, has been defined. (14)

Response 21:

MP instead of TV>

Dear reviewer, thank you for your suggestion. In accordance with your suggestion, the relevant word has been corrected as follows:

…or low TV with a higher RR (risk of excessive MP)?

Response 22:

values>

Dear reviewer, thank you for your suggestion. In accordance with your suggestion, the relevant word has been corrected as follows:

…PEEP values (i.e. lowest driving pressure for the same TV)  followed by an ARM and low TV.

Response 23:

computed> 

Dear reviewer, thank you for your suggestion. The related word has been deleted after the changes made.

Reviewer 2 Report

Comments and Suggestions for Authors

Thank you for inviting me tor review this important paper.

I have several proposals

1. to rewrite first part of the paper and to make usual introduction

2. to add future directions and how implementation of MPV and one lung ventilation can influence every day clinical practice

3. Conclusions must be reviewed. now its like single sentences.

4. in the manuscript is it very useful table, I think you must add some figures.

Author Response

Response to Reviewer 2 Comments

Dear reviewer, thank you for the comments you have made. We have already made the changes you requested. We think that the article has reached a better point in line with your suggestions.

Response 1:

I have several proposals

  1. to rewrite first part of the paper and to make usual introduction>

Dear reviewer, thank you for your suggestions. The change you requested in the introduction section has been made as follows.

The experience about VALI in ICU patients can be useful to deal with “one-lung ventilation associated lung injury” [2,3]. Protective lung ventilation (PLV) consists of a bundle of low tidal volume (TV), PEEP and alveolar recruitment maneuver (ARM) to achieve a low driving pressure. Hovewer, the application of PLV during OLV is associated with a series of questions (e.g. how low the TV; how high the PEEP etc). The concept of “open lung approach” (OLA) can help to solve some of these questions by “individualizing” the PEEP with a titration [4]. However, the beneficial effects of OLA during OLV also remain controversial. 

Response 2:

<2. to add future directions and how implementation of MPV and one lung ventilation can influence every day clinical practice>

Dear reviewer, thank you for your suggestions. The section you requested has been added as a part with the suggestion you have given.

Future directions and clinical implications;

In the future, mechanical power (MP) and personalized ventilation strategies are expected to play a crucial role in optimizing OLV during thoracic surgery. MP and “ergotrauma” are relative recent concepts:

* The respiratory rate is taken also into account of VALI

* PEEP per se is considered as a part of MP.

These principles can again change the paradigm of protective ventilation.

In any case, it has become crucial that every physician applying mechanical ventilation has to be aware of the fact that she/he has to “protect” the lung against the possible hazards of this “unphysiologic” (RIP: Hedenstierna) approach (or better: against her/his hazardous application (RIP: Gattinoni).

The shift toward data-driven, real time monitoring will allow for more precise adjustments to ventilator settings, reducing the risk of complications such as postoperative pulmonary dysfunction. As technology evolves, it is likely that these innovations will become routine in clinical practice, enhancing safety and recovery times for patients undergoing high-risk thoracic procedures.

Response 3:

<3. Conclusions must be reviewed. now its like single sentences.>

Dear reviewer, thank you for your suggestions. With the suggestion you have given, we have rewritten the conclusion part as follows:

In conclusion, OLV presents significant challenges in thoracic anesthesia, primarily in preventing hypoxemia and postoperative pulmonary complications. Recent advancements in protective ventilation strategies, including the use of low tidal volumes, optimal PEEP, and ARM when appropriate, have demonstrated promising outcomes in addressing both of these challenges. However, the effectiveness of these strategies relies on the precise application and individualized adjustment of each component: a TV of 4-6 ml/kg appears to be appropriate, but is effective only with rational application of other components. Regarding PEEP and ARM, there are still important questions to be answered in order to set new paradigms.  Notably, further investigation is needed to clarify the optimal PEEP levels and timing for ARM to fully understand their role in minimizing postoperative pulmonary complications. Thus, a tailored approach, integrating these components, remains crucial to reducing postoperative pulmonary complications and improving recovery outcomes.

Response 4:

<4. in the manuscript is it very useful table, I think you must add some figures.>

Dear reviewer, thank you for your suggestions. In accordance with your suggestions, an additional figure and table related to PEEP titration have been added to the article.

Figure1.

“Decremental PEEP titration” to achieve the lowest driving pressure according the static compliance.

The compliance curve has a sigmoid shape. The area with a steeper slope has a better compliance (i.e. less pressure is required for the same volume). The aim is to ventilate the lungs within this area, not below the lower inflection point (LIP), which would lead to atelectasis and cyclic opening-closing of the alveoli; and not above the upper inflection point (UIP), which would mean overinflation and lead to baro- volutrauma. 

A: An ARM is applied to open all the atelectatic (and recruitable) alveoli, then the ventilation is started with a high PEEP. In this region, compliance is not expected to be high.

B: Stepwise decrease of PEEP; a stepwise increase in static compliance is expected. This increase would continue until some alveoli become atelectatic.

C: When the compliance decreases again, which would mean that some alveoli become atelectatic (below LIP).

The PEEP level achieving the highest compliance is the individualized best PEEP; a new ARM has now to be performed to open the atelectatic alveoli again; and the patient is ventilated with the PEEP of best compliance.

Note: 1.  This method should be done ideally with the static compliance (with “zero-flow”), which is very difficult during the operation, some compromise (like decreasing the respiratory rate, changing the I: E ratio) be necessary. 2. Distribution of ventilation within the ventilated lung is not homogenous; therefore it should be kept in mind that even with the “individualized” PEEP, some (upper) regions are overinflated, as some (lower) are in risk of atelectasis. 

Table 2:  Some possible methods of PEEP titration. Historical methods like PEEP-FiO2 scale or “best DO2) are not included to the table; likewise CT-scan is impossible to apply during the operation in the operating room. (OR: operating room; VCV: volume-controlled ventilation; RR: respiratory rate)

Method

Principle

Comment/Concern

Driving Pressure (DP) guidance

PEEP to achieve the highest static compliance (see also Figure 1)

Static compliance (i.e. no inspiratory flow) is difficult to measure in daily practice.

Titrated PEEP does not eliminate that some (lower) areas remain atelectatic, as some (upper) areas are overinflated.

Electric impedance tomography (EIT)

Visual detection of atelectasis. PEEP to avoid atelectasis.

Difficult in OR , esp. in thoracic patients.

Lung ultrasound

Visual detection of atelectasis

Less difficult than EIT, but still difficult during the operation.
Subjective evaluation possible

Ventilatory stress index

Analysis of the slope of the pressure-time curve during VCV: PEEP to achieve a linear slope (not concave or convex)

Although being a very appropriate method, and easy at the same time, it is not used often in daily practice.

Transpulmonary pressure

Oesophageal pressure as the surrogate. The most direct way to measure the stress and strain

Specific motors necessary. Artefacts possible.

Mechanical power

Measurement takes the RR and PEEP also in account

Relative new method. Almost no experience for PEEP titration in OR.

Round 2

Reviewer 1 Report

Comments and Suggestions for Authors

The paper has been improved but still needs some revision.

My major comment refers to the legend of figure 1 and table 2. Static compliance is calculated easily with the application of an end-inspiratory hold for 3 seconds, which is provided to any ventilator. What is difficult to obtain is the pressure-volume curve, i.e. the static compliance curve, in order to identify LIP and UIP and set PEEP slightly above LIP (this is another method to assess optimal PEEP, which has been abandonned nowadays because of its complexity). Therefore, the note 1 of the figure (line 318) should be replaced by "the construction of P-V curve to identify LIP and UIP..." and in table 2 "DP guidance" should be replaced by "identification of LIP and UIP on the P-V curve"

Minor comments:

  1. line 76 increased instead of increases
  2. line 79 V/Q mismatch 
  3. line 77 is still perfused instead of ventilated
  4. line 83 the abbreviation HPV should be written in ful 
  5. line 85 add "mitigating V/Q mismatch"
  6. line 134 omit "there is"
  7. line 159 add "driving pressure of 13-14cmH2O"
  8. line 160 omit "...of 20cmH2Ofor 15-20 seconds" because there are various ways to perform ARMs
  9.  lines 157-8 There are robust experimental and clinical data that support protective ventilation, so it is not "exclusively empiric"
  10. line 214 While instead of Αs
  11. line 219 Article 37: applied before OLV initiation
  12. line 220 Article 38 is a meta-analysis
  13. line 269 add using the P-V curve
  14. line 284 6% versus 15%
  15. lines 290-291and the figure should move to line 269

Author Response

Response to Reviewer 1 Comments

Dear reviewer, thank you again for your comments.

Response 1:

curve, in order to identify LIP and UIP and set PEEP slightly above LIP (this is another method to assess optimal PEEP, which has been abandonned nowadays because of its complexity). Therefore, the note 1 of the figure (line 318) should be replaced by "the construction of P-V curve to identify LIP and UIP..." and in table 2 "DP guidance" should be replaced by "identification of LIP and UIP on the P-V curve">

Dear reviewer, thank you for your suggestion. In accordance with your suggestion, the relevant sections has been revised as follows:

Figure 1: “Decremental PEEP titration” to achieve the lowest driving pressure according the static compliance.

The compliance curve has a sigmoid shape. The area with a steeper slope has a better compliance (i.e. less pressure is required for the same volume). The aim is to ventilate the lungs within this area, not below the lower inflection point (LIP), which would lead to atelectasis and cyclic opening-closing of the alveoli; and not above the upper inflection point (UIP), which would mean overinflation and lead to baro- volutrauma.  A: An ARM is applied to open all the atelectatic (and recruitable) alveoli, then the ventilation is started with a high PEEP. In this region, compliance is not expected to be high. B: Stepwise decrease of PEEP; a stepwise increase in static compliance is expected. This increase would continue until some alveoli become atelectatic. C: When the compliance decreases again, which would mean that some alveoli become atelectatic (below LIP). The PEEP level achieving the highest compliance is the individualized best PEEP; a new ARM has now to be performed to open the atelectatic alveoli again; and the patient is ventilated with the PEEP of best compliance. Note: 1.  This method should be done ideally with the construction of P-V curve to identify LIP and UIP. 2. Distribution of ventilation within the ventilated lung is not homogenous; therefore it should be kept in mind that even with the “individualized” PEEP, some (upper) regions are overinflated, as some (lower) are in risk of atelectasis.

Table 2:  Some possible methods of PEEP titration. Historical methods like PEEP-FiO2 scale or “best DO2) are not included to the table; likewise CT-scan is impossible to apply during the operation in the operating room. (OR: operating room; VCV: volume-controlled ventilation; RR: respiratory rate)

Method

Principle

Comment/Concern

Identification of LIP and UIP on the P-V curve

PEEP to achieve the highest static compliance (see also Figure 1)

Static compliance (i.e. no inspiratory flow) is difficult to measure in daily practice.

Titrated PEEP does not eliminate that some (lower) areas remain atelectatic, as some (upper) areas are overinflated.

Electric impedance tomography (EIT)

Visual detection of atelectasis. PEEP to avoid atelectasis.

Difficult in OR , esp. in thoracic patients.

Lung ultrasound

Visual detection of atelectasis

Less difficult than EIT, but still difficult during the operation.
Subjective evaluation possible

Ventilatory stress index

Analysis of the slope of the pressure-time curve during VCV: PEEP to achieve a linear slope (not concave or convex)

Although being a very appropriate method, and easy at the same time, it is not used often in daily practice.

Transpulmonary pressure

Oesophageal pressure as the surrogate. The most direct way to measure the stress and strain

Specific motors necessary. Artefacts possible.

Mechanical power

Measurement takes the RR and PEEP also in account

Relative new method. Almost no experience for PEEP titration in OR.

Response 2:

 instead of increases>

Dear reviewer, thank you for your suggestion. In accordance with your suggestion, the relevant word has been corrected as follows:

...increased shunt (non-ventilated lung is still ventilated…

Response 3:

Dear reviewer, thank you for your suggestion. In accordance with your suggestion, the relevant word has been corrected as follows:

…decrease the V/Q mismatch.

Response 4:

perfused instead of ventilated>

Dear reviewer, thank you for your suggestion. In accordance with your suggestion, the relevant word has been corrected as follows:

…is still perfused) and…

Response 5:

Dear reviewer, thank you for your suggestion. In accordance with your suggestion, the relevant word has been corrected as follows:

…. hypoxic pulmonary vasoconstriction (HPV):

Response 6:

Dear reviewer, thank you for your suggestion. In accordance with your suggestion, the relevant part has been added as follows:

HPV (the name is self-explaining) denotes the physiological phenomenon whereby pulmonary arterioles undergo constriction in response to alveolar hypoxia, and it mitigates V/Q mismatch.

Response 7:

Dear reviewer, thank you for your suggestion. In accordance with your suggestion, the relevant word has been deleted as follows:

The last decade, a fifth trauma has been proposed namely ergotrauma…

Response 8:

Dear reviewer, thank you for your suggestion. In accordance with your suggestion, the relevant word has been added as follows:

Therefore, current recommendation appears to be rational, but is exclusively empiric: low tidal volumes of 5-6 ml/kg based on ideal body weight, positive end-expiratory pressure of 5 cmH2O / or “individualized” PEEP, plateau pressure of less than 25 cmH2O, driving pressure of 13-14 cmH2O and alveolar recruitment maneuvers (ARM) (only if necessary) of 20 cmH2O for 15 to 20 seconds [23-25].

Response 9:

Dear reviewer, thank you for your suggestion. In accordance with your suggestion, the relevant word has been deleted as follows:

Therefore, current recommendation appears to be rational, but is exclusively empiric: low tidal volumes of 5-6 ml/kg based on ideal body weight, positive end-expiratory pressure of 5 cmH2O / or “individualized” PEEP, plateau pressure of less than 25 cmH2O, driving pressure of 13-14 cmH2O and alveolar recruitment maneuvers (ARM) (only if necessary) [23-25].

Response 10:

Dear reviewer, thank you for your suggestion. In accordance with your suggestion, the relevant sentence  has been corrected as follows:

All the components of protective ventilation are associated with unanswered questions, these components can be discussed in details.

Response 11:

While instead of Αs>

Dear reviewer, thank you for your suggestion. In accordance with your suggestion, the relevant word  has been corrected as follows:

While the majority of the studies in thoracic surgery have shown that recruitment maneuvers reduce both postoperative complications and improve oxygenation [32-34],  some studies have shown that ARM has no effect or can be associated even with an increase in mortality [35,36].

Response 12:

before OLV initiation>

Dear reviewer, thank you for your suggestion. In accordance with your suggestion, the relevant word  has been corrected as follows:

Various studies have demonstrated that recruitment maneuvers during one lung ventilation are more beneficial compared to maneuvers applied before OLV initiation [38].

Response 13:

Dear reviewer, thank you for your suggestion. In accordance with your suggestion, the relevant word  has been corrected as follows:

The meta-analysis related with ARMs has demonstrated that recruitment maneuvers during one lung ventilation are more beneficial compared to maneuvers applied before OLV initiation [38].

Response 14:

using the P-V curve>

Dear reviewer, thank you for your suggestion. In accordance with your suggestion, the relevant word  has been added as follows:

The drawbacks of “fixed” PEEP application can be solved with “individualization” of PEEP via different methods of titration. The "Open Lung approach" (OLA), proposed for ARDS patients, uses maximal alveolar recruitment followed by PEEP titration using the pressure-volume curve. Several studies have found that OLA during thoracic surgery improved oxygenation, lung compliance, and transpulmonary pressure (PL), without significant hemodynamic effects, and indicated that PL is crucial in understanding lung and chest wall mechanics during one-lung ventilation [25,49-51].

Response 15:

15%>

Dear reviewer, thank you for your suggestion. In accordance with your suggestion, the relevant word  has been added as follows:

Among 1308 patients, severe postoperative pulmonary complications were lower in the OLA group (6% versus 15%) than in the standard ventilation group [4]. 

Response 16:

Dear reviewer, thank you for your suggestion. In accordance with your suggestion, the relevant part  has been revised as follows:

The drawbacks of “fixed” PEEP application can be solved with “individualization” of PEEP via different methods of titration. The "Open Lung approach" (OLA), proposed for ARDS patients, uses maximal alveolar recruitment followed by PEEP titration using the pressure-volume curve. The rationale of OLA is to target a decrease of driving pressure (DP), i.e. a lower pressure for the same tidal volume, or better to say a better static compliance. (Figure 1). Several studies have found that OLA during thoracic surgery improved oxygenation, lung compliance, and transpulmonary pressure (PL), without significant hemodynamic effects, and indicated that PL is crucial in understanding lung and chest wall mechanics during one-lung ventilation [25,49-51].
